Optimization and action mechanism of pollutant removal performance of unsaturated vertical flow constructed wetland (UVFCW) driven by substained-release carbon source

Wang Yuechang 1 2
Zhuang Lin-Lan 1
http://orcid.org/0009-0000-1261-039X Liu Shuang 3
Su Yuanjun 3
Hu Zhen 1 huzhen885@sdu.edu.cn
Zhang Jian 1 zhangjian00@sdu.edu.cn
Wang Xiaoping 2
Cui Shaoming 2
Peng Gang 2
Xie Shuting 3
1 Shandong Key Laboratory of Water Pollution Control and Resource Reuse, School of Environmental Science & Engineering, Shandong University , Qingdao , China
2 Beijing Further Tide Eco-Construction Co., Ltd , Beijing , China
3 Further Tide Eco-Construction (Hubei) Co., Ltd , Wuhan , China
Phairuang Worradorn
Electronic publication date: 2025 Jan 16
Publication date: 2025
Volume: 13
Electronic Location ID: e18819
Received 2024 Oct 21; Accepted 2024 Dec 15
Copyright: © 2025 Wang et al.
Copyright year: 2025
Copyright holder: Wang et al.
License: This is an open access article distributed under the terms of the Creative Commons Attribution License, which permits unrestricted use, distribution, reproduction and adaptation in any medium and for any purpose provided that it is properly attributed. For attribution, the original author(s), title, publication source (PeerJ) and either DOI or URL of the article must be cited.
License URL: https://creativecommons.org/licenses/by/4.0/

Keywords: UVFCW, Sustained-release carbon source, Nitrogen removal, Carbon source placement method, Denitrification

Funding: National Science Foundation of China 51925803 National Key Research and Development Program of China 2021YFC3200602 Beijing Further Tide Eco-construction Co., Ltd 2023YLCRD001 This research was supported by the National Science Foundation of China (No. 51925803), National Key Research and Development Program of China (No. 2021YFC3200602) and Beijing Further Tide Eco-construction Co., Ltd (2023YLCRD001). The funders had no role in study design, data collection and analysis, decision to publish, or preparation of the manuscript.

==============================
Constructed wetland (CW) technology has attracted much attention due to its economical and environmentally friendly features. The low dissolved oxygen (DO) and low carbon/nitrogen (C/N) ratio in the wetland influent water affect the treatment performance of CW, resulting in a decrease in the removal efficiency of ammonia nitrogen (NH4+-N) and nitrate nitrogen (NO3−-N). In order to address this problem, this study optimized the pollutants removal performance of unsaturated vertical flow constructed wetland (UVFCW) by adding sustained-release carbon sources (corn cobs + polybutylene adipate terephthalate (PBAT)). The results showed that the sustained-release of carbon source increased the carbon source in UVFCW, thus increasing the abundance and activity of denitrifying microorganisms and enhancing the denitrification reaction, ultimately improving the removal of NO3−-N, with its removal efficiency reaching up to 95.50%. The placement method of sustained-release carbon source mainly affected the distribution of carbon source and DO in water body, thus influencing the relative abundance of microorganisms, finally affecting the removal of pollutants. Among them, the removal efficiency of total nitrogen (TN), NO3−-N, and total phosphorus (TP), and the relative abundance of denitrifying microorganisms in the CWR-Cu (uniform placement of sustained-release carbon source) were significantly higher than those in the CWR-Ca (centralized placement above) and CWR-Cb (centralized placement below) (p < 0.05). The surface C:O (carbon:oxygen) ratio of sustained-release carbon source after water treatment showed a decreasing trend, and CWR-Cu exhibited the greatest decrease in C:O ratio. In summary, CWR-Cu achieved the highest utilization of the carbon source and produced the largest number of heterotrophic microorganisms. This study reveals that CWR-Cu is a structural process for the efficient removal of nitrogen and phosphorus pollutants, and our findings provide theoretical basis and technical support for actual projects.

Introduction

Constructed wetlands (CW) are widely used as an economical and environmentally friendly water treatment technology. CW mainly employs substrates, plants, and microorganisms to physically, chemically, and biologically treat sewage so as to achieve the purpose of water purification. Vertical flow constructed wetlands (VFCW) are widely used due to their high treatment capacity and wide application range. Compared with traditional CW, VFCW has higher O2 input, but high O2 availability will consume a large amount of carbon source (Saeed & Sun, 2012), thus limiting denitrification reaction and reducing total nitrogen removal efficiency. To address this problem, unsaturated vertical flow constructed wetland (UVFCW) has been proposed in some studies (Pelissari et al., 2017, 2018), namely, a certain amount of water accumulation is set at the bottom to create anoxic, anaerobic, or aerobic conditions and increase the diversity of microorganisms so as to promote the removal of various pollutants. Many studies have shown that UVFCW can overcome the defects of VFCW promote denitrification reaction, and improve the removal of nitrate (NO3−)-N (Pelissari et al., 2017; Sgroi et al., 2018). Substrate in CWs were exposed to air for high-efficiency oxygenation to biofilm, instead of being submerged under water. Zhou et al. (2021) showed that DO showed an increasing trend with the decrease of water depth in UVFCW. Compared to the unsaturated zone near the inlet, saturated zone near the bottom forms anaerobic condition. Therefore, the differentiated redox conditions could enhance both nitrification and denitrification processes. However, at any unsaturated rate of VFCW, microorganisms will consume carbon sources during nitrification, and the carbon source in the influent water is also limited, which severely restricts the denitrification rate and affects the removal of NO3−-N.

The removal of NO3−-N from CW relies on sufficient carbon sources to provide electron donors for denitrification (Sun et al., 2010). The introduction of external carbon sources into CW can significantly improve the removal efficiency of NO3−-N (Gu et al., 2021). Solid carbon sources have become a research hot spot since they can overcome multiple shortcomings of liquid sustained-release carbon sources such as easy loss, difficulty in controlling the amount of addition, high transportation costs and operation costs (Sun et al., 2022). Li et al. (2024) added straw to VFCW and increased total nitrogen (TN) removal efficiency from 36.0% to 44.9%, but straw addition caused ammonia pollution. Yang et al. (2018) used polyhydroxybutyrate-co-valerate (PHBV) and polycaprolactone (PCL) as sustained-release carbon sources in VFCW, and increased the TN removal efficiency up to 97.03%. However, these polymers have such shortcomings as easy dissolution, unstable carbon release, and low carrier strength. All the above-mentioned sustained-release carbon sources have been tested in VFCW, but the oxygen is more sufficient in UVFCW than in VFCW, and thus the effectiveness of the sustained-release carbon source in UVFCW remains to be further explored.

Zou et al. (2024) added corn cobs as a carbon source in the middle of the VFCW, and increased the NO3−-N removal efficiency from the system to 98.38%. Zhang et al. (2023) added natural plant materials as a carbon source in the upper part of the VFCW, and increased the TN removal efficiency from the effluent water by 49.62–61.3%. The above studies indicated that adding carbon sources at different locations will affect the removal of pollutants. The dissolved oxygen (DO) in the UVFCW water body tends to decrease with depth, which will cause the different degrees of oxygen (O2) and carbon source consumption by microorganisms along the water flow (Zheng et al., 2020). Currently, little research has been done on the mechanism of pollutant removal improvement by optimizing the placement of carbon sources in UVFCW, and thus it is necessary to investigate the effects of carbon source placement on pollutant removal.

Considering this, the current study improved the nitrogen removal efficiency by optimizing the addition method of sustained-release carbon source (SRCS) in UVFCW, and examined the morphological changes of sustained-release carbon source before and after the experiment and the differences in the distribution of microbial communities in the substrate, with an aim to reveal the mechanism underlying denitrification performance improvement. Our findings will provide a theoretical basis and technical support for the related engineering applications.

Materials and Methods

UVFCW

Zhuang, Song & Zhang (2020) used hydrophilic material, i.e., cotton micro-fiber with the total length of 4.5 m was spiral filled in the pores of gravels in the tidal unsaturated area of CWTU (a tidal unsaturated constructed wetland) to extend the hydraulic retention time and enhance the removal efficiency of ammonia nitrogen (NH4+-N). In light of this, the present study added water-conducting materials (rope-shaped artificial water grass, CWR) to the unsaturated zone to enhance the removal of NH4+-N. Besides, carbon sources were mixed with the common substrate and filled in different zones of CWs according to the carbon source placement method as follows. As shown in Fig. 1, four types of unsaturated constructed wetlands were designed in this study, namely, CWR (Fig. 1A), CWR+Cu (carbon source were mixed with the common substrate uniformly, Fig. 1B), CWR+Ca (carbon source was concentrated at the depth of three-quarters from the bottom of the saturation zone, Fig. 1C) and CWR+Cb (carbon source was concentrated in the bottom quarter of the saturation zone, Fig. 1D). The actual UVFCW was shown in Figs. 1E and 1F. In this article, the sustained-release carbon source is made by mixing polybutylene adipate terephthalate (PBAT) and corncob for long-term and high-efficiency application (Zhang et al., 2024). The purpose of using sustained-release carbon sources is to continuously release carbon into constructed wetlands to enhance denitrification. We investigated the characteristics of corncob and PBAT. Although corncob has a good carbon-release effect, its release rate was too fast, and the duration of carbon release is not lasting. In contrast, the release rate of carbon by PBAT was slow, which could encapsulate corncob, slowing down the release of carbon from corncob and increasing the its persistence. Previous studies have shown that the persistence of corncob encapsulated by PBAT was about 85 days longer than that of unencapsulated corncob, and the cost was 7 yuan per kg lower than that of pure PBAT.

Figure 1 Structural optimization of CWR (A), CWR+Cu (B), CWR+Ca (C), CWR+Cb (D), and UVFCW laboratory actual pictures (E) and (F).

CWR (CW with rope-shaped artificial water grass, (A)), CWR+Cu (carbon source were mixed with the common substrate uniformly, (B)), CWR+Ca (carbon source was concentrated at the depth of three-quarters from the bottom of the saturation zone, (C)) and CWR+Cb (carbon source was concentrated in the bottom quarter of the saturation zone, (D)). (E) Represents the actual UVFCW picture, (F) controls the unsaturated ratio of UVFCWd.

The cylindrical UVFCW was made of acrylic material, and the chemical oxygen demand (COD), TP, NH4+-N, and NO3−-N in the influent water were set as 50, 0.5, 5, and 10 mg/L, respectively. According to the previously reported method (Zhuang, Song & Zhang, 2020), the trace elements such as calcium chloride (CaCl2), magnesium sulfate (MgSO4), copper sulfate (CuSO4), molybdic acid (H2MoO4), and boric acid (H3BO3) were added, and the conventional elements, trace elements, and deionized water were mixed at certain concentrations and placed in large plastic barrels to simulate wastewater in wastewater treatment plant. The influent water flow rate of each UVFCW was 4.17 mL/min; the hydraulic load was 0.34 m3/m2·d; the hydraulic retention time (HRT) was 24 h; and the substrate was 3–5 mm gravel. A pierced polyethylene plastic column was placed in the middle of the substrate to monitor the DO of the water body. Considering the low contribution of plants to the pollutant removal from VFCW, no plants were set in this study (Zhao et al., 2011).

Sample collection and analysis

Pollutant analysis method

First, the microorganism growth test on the substrate was conducted on the UVFCW. Each UVFCW was sampled every 3 days until the pollutant removal efficiency reached a stable level, and at 3 days before the experiment was finished, water samples were taken every day. Water samples were taken along the reactor at intervals of 10 cm with three replicates (namely, three samplings) to determine DO, COD, TP, NH4+-N, NO3−-N, and TN. Specifically, DO was determined by a portable dissolved oxygen detector, COD by a rapid digestion method, TP by molybdate spectrophotometry, NH4+-N by Nessler’s reagent spectrophotometry, NO3−-N by a spectrophotometer, and TN by a spectrophotometer.

Material characterization

The sustained-release carbon source was collected after the test, and the prepared sustained-release carbon source (corn cob + PBAT) and the samples before and after use were tested using scanning electron microscope (JSM. 6380LV; SEM, Tokyo, Japan) and an energy dispersive X-ray spectrometry (IE350; EDS, Birmingham, UK) to obtain the information on their appearance characteristics, surface configuration, and the surface element content and distribution. Before the SEM test, the block sample was treated with gold spraying or carbon spraying, with a normal acceleration voltage of 20 kV and a maximum of 30 kV. Before the EDS test, the sample was pre-treated with liquid nitrogen.

High-throughput sequencing

After the experiment, the materials in the treatment group were mixed and sampled. The mixed samples were stored in a −80 °C environment in a ziplock bag. The high-throughput sequencing of the collected samples was performed by Shanghai Paisenor Co., Ltd. (Shanghai, China). The primers in the bacterial 16S rRNA V3-V4 region were selected on the Illumina MiSeq system to test and analyze the community structure of the samples. The forward and backward primers were 338 F (sequence: 5′-ACTCCTACGGGAGGCAGCAG-3′) and 806 R (sequence: 5′-GGACTACHVGGGTWTCTAAT-3′) (Claesson et al., 2009). The microbial community in the samples was analyzed on the Paisenor Cloud Platform of Shanghai Paisenor Co., Ltd. (Shanghai, China).

Statistical analysis

The data obtained from the experiment were processed using R (version 4.02). The Origin 9 (version 2019b) was used to plot bar graphs of the pollutant concentration in influent and effluent water and pollutant removal efficiency. The “circlize” package in R (version 4.3.2) was employed to plot the abundance chord diagram at the microbial phylum level. The “corrplot”, “ggplot2”, “reshape2”, “psych”, and “pheatmap” packages in R were used to perform inter-group correlation analysis and plot the correlation between environmental parameters and the relative abundances at phylum and genus levels (p < 0.05). The “pheatmap” package was used for plotting heatmap and analyzing the species differences between samples. The “ggplot2” package was used to draw principal component analysis (PCA) diagrams to quantitatively display the degree of difference in sample species composition.

Results and discussion

Adding sustained-release carbon source enhances nitrogen and phosphorus removal

In this study, CWs with different unsaturation rates were investigated. With the changes of unsaturated zone, the DO and pH along water pathway also changed (Supplemental Material). Moreover, the various redox condition led to different TN removal. The results indicated that when the unsaturation rate was 50%, COD and TP removal efficiency were desirable, but the removal effects on TN and NO3−-N were poor, whose removal efficiency were both less than 25%. In addition, the C/N ratio of influent water was low (3.5). Based on these results, we speculated that the lack of carbon source in the denitrification zone was the main factor affecting the removal efficiency of TN and NO3−-N.

In order to increase the removal efficiency of TN and NO3−-N, CW50% was improved, as shown in Fig. 1. As shown in Fig. 2A, the COD removal efficiency in CWR control group was higher than that of all the sustained-release carbon source treatment groups. The possible reason was that the amount of organic matter released by the sustained-release carbon source was greater than that from aerobic microbial degradation, resulting in excessive organic matter discharge from the system with water (Huang et al., 2020). The COD removal efficiency in CWR-Cu group was the lowest. As shown in Fig. 2B, the removal efficiency of NH4+-N in CWR group was higher than that in the sustained-release carbon source treatment groups. Microbial ammonia oxidation is the first step to remove NH4+-N from TN. The removal of NH4+-N mainly depends on the nitrification reaction under aerobic conditions (Li et al., 2019; Vymazal, 2007). It can be inferred that heterotrophic microorganisms will compete for oxygen during degradation of COD and NH4+-N (Li et al., 2018), resulting in a downward trend in the removal of NH4+-N in the sustained-release carbon source treatment groups. Among them, the removal efficiency of NH4+-N in the CWR-Cb group was the highest, and the removal efficiency of CWR-Cu was the lowest. As shown in Fig. 2A, the COD concentration in CWR-Cu was the highest, indicating that the high concentration of COD inhibited the removal of NH4+-N.

Figure 2 Effluent water quality along with operation time.

The removal efficiency of NO3−-N and TN in the sustained-release carbon source treatment groups were higher than those in CWR group (Figs. 2C and 2D), indicating that the addition of sustained-release carbon source significantly increased NO3−-N and TN removal efficiency. Our results were consistent with one previous report that adding sustained-release carbon source (calamus + polyhydroxybutyrate-co-valerate (PHBV)) to CW resulted in the effective removal of NO3−-N and TN (Li et al., 2024). Our data also showed that CWR-Cu treatment exhibited the best removal effect on NO3−-N and TN, which was better than CWR-Ca and CWR-Cb treatments, indicating that the placement of the sustained-release carbon source had a significant impact on the removal of pollutants. Zhou et al. (2022) employed sucrose and reed litter as VFCW carbon sources, and TN removal efficiency in their study was 84.9%; Sun et al. (2022) utilized calamus as the sustained-release carbon source in CW, and the NO3−-N removal efficiency in their study was 76.03%; and the removal efficiency of NO3−-N and TN in these two studies were significantly lower than those in our CWR-Cu treatment, indicating that the sustained-release carbon source in this study had excellent denitrification function.

The removal efficiency of TP in the sustained-release carbon source treatment groups was significantly higher than that of CWR control group (Fig. 2E), indicating that the addition of sustained-release carbon source can promote the removal of TP. Since the UVFCW substrate was the same, we speculated that with the increasing organic matter concentrations in UVFCW, the adsorption sites of PO4−-P were increased in the sustained-release carbon source treatment group, thus improving the removal efficiency of TP (Liu & Hu, 2019). The removal efficiency of TP in CWR-Cu treatment group (81.87%) was better than that in CWR-Ca (69.74%) and CWR-Cb (58.06%) groups, indicating that the placement of the sustained-release carbon source had a significant impact on the removal of TP. The uniform placement of sustained-release carbon source exhibited the best effect on TP removal.

In summary, the sustained-release carbon source treatment can significantly improve the removal efficiency of NO3−-N, TN, and TP. Different placement methods of sustained-release carbon sources exhibit different removal effects on pollutants. Next, we further investigated the purification effect of the different sustained-release carbon source placement methods on pollutants.

Effects of carbon source placement on pollutant removal

To understand the mechanism of pollutant removal by carbon source placements, we investigated the water quality changes with water flow in UVFCW. As shown in Fig. 3, the COD removal efficiency in CWR control group along the reactor was higher than that of the sustained-release carbon source treatment groups (Fig. 3A). The COD removal efficiency in CWR-Cu group decreased from 36.58% to −3.53% with the increasing CW depth, and that in CWR-Ca and CWR-Cb treatment groups were the lowest at the sustained-release carbon source placement point (50 and 70 cm from the top of the column), with a COD removal efficiency of −362.80% and −194.73%, respectively. The possible reason might be that the sustained-release carbon source under CWR-Cu treatment was more widely distributed in space, and the released carbon source had the highest contact time with bacteria for denitrification, resulting in a downward trend in COD removal efficiency in CWR-Cu. The sustained-release carbon source under CWR-Ca and CWR-Cb treatments released a large amount of organic carbon locally, causing excessively high COD concentration for heterotrophic bacteria, lowering the efficiency for denitrification consumption. The above results jointly indicated that the placement method of sustained-release carbon source affected the carbon release amount, and that the sustained-release effect of uniformly-placed carbon source was better than that of concentratedly-placed carbon source.

Figure 3 Water quality changes along with water flow direction.

The NH4+-N removal efficiency (100%) in CWR control group along reactor (50, 60, 70, and 80 cm from column top) was higher than that in the groups added with sustained-release carbon source (Fig. 3B). The NH4+-N removal efficiency under CWR-Ca and CWR-Cb treatments was the lowest at the sustained-release carbon source placement point (50 and 70 cm), which was 77.42% and 88.42%, respectively. The NH4+-N removal efficiency in CWR-Cu group showed a decreasing trend (from 99.54% to 87.56%) along the reactor (from 50 to 80 cm), indicating that the corn cobs as sustained-release carbon source might have produced NH4+-N, potentially inducing ammonia pollution (Bao et al., 2016). At the same time, microorganisms will compete for oxygen during the degradation of COD and NH4+-N, further reducing the removal of NH4+-N. The removal efficiency of NH4+-N in CWR-Cb group was generally higher than that in CWR-Cu and CWR-Ca groups. As shown in Fig. 3F, the DO in CWR-Cu and CWR-Ca groups was 0–1 mg/L, and that in CWR-Cb was 1.5–3.5 mg/L, indicating that the placement of the sustained-release carbon source affected the DO removal from the entire system. It has been reported that DO < 0.5 mg/L can inhibit the nitrification process in CW (Tao et al., 2021), based on which, we speculated that the low DO in the uniform placement and upper concentrated placement of the sustained-release carbon source might inhibit the nitrification reaction and reduce the removal efficiency of NH4+-N, and that that in lower concentrated placement of the sustained-release carbon source would show an opposite trend.

The removal efficiency of NO3−-N and TN by CWR-Cu, CWR-Ca and CWR-Cb were higher than those by CWR (Figs. 3C and 3D), indicating that the sustained-release carbon source at different depths (from 50 to 80 cm from column top) played a role and promoted the denitrification reaction. Zheng et al. (2022) used alkaline reed as an external sustained-release carbon source, and the removal efficiency of NO3−-N and TN increased from −15.97% and 53.13% to 95.5% and 93.7%, which also showed that the addition of sustained-release carbon source significantly promoted denitrification. The removal efficiency of NO3−-N and TN by CWR-Cu at all depths were higher than those by CWR-Ca and CWR-Cb, among which, CWR-Cb had the poorest removal effect on NO3−-N and TN, indicating that the placement of the sustained-release carbon source affected the denitrification. It may be related to the placement of sustained-release carbon sources affecting DO and carbon source distribution. As shown in Fig. 3F, CWR-Cb was in an aerobic state, which limited its denitrification reaction, resulting in undesirable removal efficiency of NO3−-N and TN in CWR-Cb. The anaerobic environment in CWR-Cu was the best, and its COD concentration in the whole system was high (Fig. 3A), and thus the sufficient carbon source and low DO environment promoted the denitrification reaction (Hong et al., 2017), eventually enhancing the removal of NO3−-N and TN by CWR-Cu.

The changes of TP in different treatment groups at all four depths were shown in Fig. 3E. As the depth of CWs increased, the removal efficiency of TP in all treatment groups showed an upward trend. Among them, the removal effect on TP by CWR-Cu and CWR-Ca at each depth was better than that by CWR and CWR-Cb, indicating that the placement methods of the sustained-release carbon source had a significant impact on the removal of TP, namely, the uniform placement of the sustained-release carbon source and the concentrated placement at the top promoted the removal of TP, while the concentrated placement at the bottom inhibited the removal of TP. The high removal efficiency of TP might be due to the fast carbon release rate of the uniformly-distributed and upper-concentrated sustained-release carbon source, providing the carbon source for the growth and reproduction of phosphorus-accumulating bacteria. The high-concentration COD zone is conducive to the growth and reproduction of phosphorus-accumulating bacteria, thus increasing the removal efficiency of TP, which was consistent with our observation in “Effects of sustained-release carbon source placement method on nitrogen- and phosphorus-removing microorganisms” that the higher relative abundance of Pseudomonas was observed in CWR-Cu and CWR-Ca.

In summary, the sustained-release carbon source placement method mainly affects the removal of pollutants by affecting the distribution of carbon sources and DO in CWs. Our results reveal that the uniform placement of sustained-release carbon source can significantly improve the removal efficiency of pollutants. Further, we explored sustained-release denitrification mechanism by investigating sustained-release carbon source characterization.

Denitrification performance and characterization analysis of sustained-release carbon source

The denitrification rate of the groups treated with additional sustained-release carbon source was significantly higher than that of CWR control group (p < 0.05). The CWR-Cu exhibited the highest denitrification rate (4.05 g/(m3·d)) (Fig. 4A) and the highest COD concentration in its effluent water (Fig. 3A), suggesting that evenly placed sustained-release carbon sources significantly increased the organic carbon compounds in CW and promoted the microbial denitrification process (Gu et al., 2021). As shown in Fig. 4B, the density of the sustained-release carbon sources in CW group decreased, and that in CWR-Cu group was the lowest, indirectly suggesting that evenly placed sustained-release carbon sources released the most amount of organic matter during the same test period.

Figure 4 Denitrification rate by (A) and density (B) of sustained-release carbon source in CWR, CWR-Cu, CWR-Ca, CWR-Cb groups.

CWR-Cu, uniformly placed sustained-release carbon source; CWR-Ca, the upper-concentrated sustained-release carbon source; CWR-Cb, the lower-concentrated sustained-release carbon source; CK (control, CWR), the original sustained-release carbon source (before use).

The sustained-release carbon sources in different groups before and after use and their SEM and EDS analyses were in Fig. 5. The original sustained-release carbon source (before use) was brownish yellow (Fig. 5A), and the sustained-release carbon source after use in CWs was black or light yellow (Figs. 5D, 5G, and 5J), which is related to the carbon release amount of the sustained-release carbon source. The surface of the original material was coated with PBAT to form a protective layer, exhibiting the stable carbon release effect (Fig. 5B). A great amount of fibrous substances appeared on the surface of the carbon source used in CWR-Cu and CWR-Ca, and some fibrous substances appeared in CWR-Cb, which might be rod-shaped bacteria generated during water treatment (Figs. 5E, 5H, and 5K) (Shao et al., 2020). The C:O ratio on the surface of the original material was 59:41 (Fig. 5C). After use, the C:O ratio on the surface of the sustained-release carbon source decreased (Figs. 5F, 5I and 5L).

Figure 5 Sustained-release carbon source in CK (A–C) CWR-Cu (D–F), CWR-Ca (G–I), and CWR-Cb (J–L) groups before (sustained-release carbon source without water treatment) and after use and their SEM and EDS analyses.

As shown in Fig. S1, the intensity of the vibration peak of the carbonyl group (-C=O) at 1,720 cm−1 in the sustained-release carbon source after water treatment was slightly increased, compared with the original material (before use) (Fig. S1B). The carbonyl content ranked in the order of CWCu > CWCa ≈ CWCb > CWCK. The increase in carbonyl content indicated that the sample was partially degraded. The intensity of the vibration peaks of carboxyl (-COOH) and hydroxyl (-OH) at 3,400 cm−1 of the sustained-release carbon source after use increased slightly (Fig. S1C), suggesting that the hydrophilicity of the sustained-release carbon source was enhanced. The hydrophilic group content ranked in the order of CWCu > CWCa ≈ CWCb > CWCK. After water treatment, the -COOH, -OH and -C=O groups on the surface of the sustained-release carbon source increased, resulting in an increase in the oxygen content on the surface of the sustained-release carbon source (Kara et al., 2021; Montazer, Habibi Najafi & Levin, 2020). The C:O ratio on the surface of the sustained-release carbon source in CWR-Cu group exhibited the greatest decrease, indicating that the sustained-release carbon source in CWR-Cu group was most fully utilized.

Effects of sustained-release carbon source placement method on nitrogen- and phosphorus-removing microorganisms

Figure 6A exhibited the relative abundance chord diagram of microorganisms at the phylum level. Proteobacteria (11.14–59.30%) and Bacteroidota (3.89–39.98%) were the dominant phyla in the microbial communities in all the CWs. Proteobacteria is crucial in driving carbon and nitrogen cycling (Al Ali et al., 2020; Yan et al., 2023) and maintaining organic matter degradation efficiency in each CW (Man et al., 2020). The relative abundance of Proteobacteria in CWR was generally higher than that in other CWs, indicating that the organic matter released by the sustained-release carbon source inhibited the reproduction of Proteobacteria to a certain extent. Bacteroidota is a common chemically heterotrophic denitrifying bacteria that can decompose macromolecules to enhance biochemical properties (Si et al., 2018). The relative abundance of Bacteroidota in the groups treated with sustained-release carbon source was higher than that in CWR, indicating that sustained-release carbon source could increase the relative abundance of denitrifying bacteria in CW. The highest abundance (39.98%) was observed in CWR-Cu, implying that the uniform placement of sustained-release carbon source was more conducive to the reproduction of denitrifying bacteria than the centralized placement. Taken together, the addition and placement of sustained-release carbon source significantly affected the composition of microbial phyla.

Figure 6 String diagram of the relative abundance proportion of phylum-level microorganisms, (B) accumulation bar diagram of the relative abundance proportion of genus-level microorganisms, (A) represents the upper unsaturated zone of CW, and (B) represents the lower. (C) Mechanism diagram of nitrogen and phosphorus removal.

Figure 6B showed the microorganism accumulation at the genus level. Denitrifying bacteria can convert NO3− into N2 when the carbon source is sufficient (Fig. 6C) (Sun et al., 2022). The total relative abundance of the denitrifying bacteria genera Denitratisoma, Allorhizobium and Thauera was significantly higher in the sustained-release carbon source treatment groups than in CWR (0.08%) (p < 0.05), indicating that with the addition of sustained-release carbon sources, the relative abundance of denitrifying bacteria in each treatment group was also increased (Fu et al., 2017; Yu et al., 2019). The total relative abundance of denitrifying bacteria in CWR-Cu group was significantly higher than that in CWR-Ca and CWR-Cb groups, indicating that uniform placement of sustained-release carbon source could promote the reproduction of denitrifying bacteria and strengthen denitrification, compared with concentrated placement, which was consistent with the observation of the highest NO3−-N removal efficiency in CWR-Cu group (Fig. 3C). Pseudomonas belongs to denitrifying phosphorus-accumulating bacteria. Under anaerobic/anoxic conditions, this bacterial genus can use NO3− as an electron receptor to complete phosphorus uptake and denitrification through its metabolism, and it can also store phosphate present in the environment in the form of poly-P in cells (Zhang et al., 2020). In this study, the added sustained-release carbon source provided sufficient organic carbon sources for phosphorus-accumulating bacteria, allowing them to fully absorb phosphorus (Pietro & Ivanoff, 2015). The highest relative abundance of Pseudomonas (1.17%) was observed in CWR-Cu, which was in line with the highest TP removal efficiency observed in CWR-Cu group (Fig. 3E).

The α-diversity and richness of microorganisms in each treatment group were shown in Table 1. Coverage represents the reliability of sequencing. The larger the coverage value, the more reliable the sequencing results. The coverage of the eight samples was greater than 0.99, indicating that the sequencing results could fully reflect the actual situation of microorganisms in CW. Chao 1 represents the richness of species, the larger the Chao 1 value, the higher the richness; Shannon denotes the species evenness, the larger the Shannon value, the better the species evenness; and Simpson indicates the species diversity. The smaller the Simpson value, the higher the species diversity. The Chao 1 and Shannon indexes in CWR are higher than those in the sustained-release carbon source treatment groups, but the Simpson index in some treatment groups was lower than that in CWR, indicating that the addition of sustained-release carbon sources inhibited the richness and diversity of some species, but increased the diversity of other species, such as denitrifying bacteria. Chao 1 index value in uniformly placed sustained-release carbon source group was higher than that in concentratedly-placed sustained-release carbon source groups, indicating that uniform placement of sustained-release carbon sources was conducive to increasing microbial species richness.

Table 1 Microbial α-diversity and richness in different groups.

Sample	Coverage	Chao1	Shannon	Simpson	
CWR	0.99	3,496.75	8.55	0.97	
CWR-Cu	1	2,777.09	7.63	0.95	
CWR-Ca-A	1	2,054.21	7.81	0.98	
CWR-Ca-B	1	2,274.51	7.56	0.9	
CWR-Cb	1	2,223.49	7.97	0.98	

Figure 7A showed the correlation heatmap between pollutant removal efficiency and the microbial relative abundance at phylum level. The heatmap showed that Bacteroidota had a significant positive correlation with the removal efficiency of TN and NO3−-N. The higher the relative abundance of Bacteroidota, the higher the removal efficiency of TN and NO3−-N, indicating the denitrification function of Bacteroidota (Si et al., 2018). Figure 7B shows the correlation heatmap between pollutant removal efficiency and the microbial relative abundance at genus level. The results showed that denitrifying bacteria such as Denitratisoma, Allorhizobium, and Thauera were significantly negatively correlated with NH4+-N removal efficiency (p < 0.05), but positively correlated with the removal efficiency of TN and NH4+-N, indicating that the higher the relative abundance of denitrifying bacteria, the higher the removal efficiency of TN and NO3−-N, but excessively high NH4+-N concentration had a certain inhibitory effect on denitrifying bacteria.

Figure 7 Correlation heatmap between microbial relative abundance at phylum (A) and genus (B) levels and removal efficiency of pollutants as well as DO in UVFCW.

* represents ≤ 0.05, ** represents ≤ 0.01, *** represents ≤ 0.001.

The heatmap of species composition at the phylum level was shown in Fig. 8A. There were large differences at microbial phylum level composition between CWR and the sustained-release carbon source treatment groups, indicating that the addition of sustained-release carbon source changed the phylum-level microbial composition in the system. The phylum-level microorganisms in CWR group were mainly positively correlated with Acidobacteriota, Chloroflexi, and Myxococcota, and those in the sustained-release carbon source treatment groups were mainly positively correlated with Desulfobacterota and Firmicutes. The heatmap of species composition at the genus level was shown in Fig. 8B. Consistent with the phylum-level observation, the addition of sustained-release carbon source changed the types of genus-level microorganisms in the system. The genus-level microorganisms in CWR-Ca-A and CWR-Cb had similar compositions and were positively correlated with denitrifying microorganisms (such as Denitratisoma). The two axes of PCA jointly explained 68.7% of the relationship between species information changes and the environment (Fig. 8C), among which CWR-Ca-A and CWR-Cb were close on the PCo1 axis (50.1%), which was consistent with the observations on Figs. 8A and 8B, suggesting that CWR-Ca-A and CWR-Cb were similar in species composition.

Figure 8 Heatmap of species composition and PCA (C) analysis of functional units at phylum (A) and genus (B) levels.

(A) represents the upper unsaturated zone of CW, and (B) indicates the lower saturated zone of CW.

Conclusion

This study showed that by adding sustained-release carbon source to the UVFCW saturation zone, the relative abundance of denitrifying microorganisms (Denitratisoma, Allorhizobium and Thauera) was increased, then enhancing the denitrification reaction, eventually increasing removal efficiency of NO3−-N and TN. The placement of the sustained-release carbon source mainly affected the distribution of carbon sources in the water body and DO, thus influencing the relative abundance of microorganisms, eventually affecting the pollutant removal. The removal efficiency of TN (91.12%), NO3−-N (95.50%), and TP (81.87%), the relative abundance of denitrifying microorganisms, and the denitrification rate (4.05 g/(m3·d)) in CWR-Cu group with uniformly-placed sustained-release carbon source were higher than those in CWR-Ca and CWR-Cb groups with concentratedly placed sustained-release carbon source. The results of SEM and EDS analyses of the sustained-release carbon source showed that after water treatment, the density of the sustained-release carbon source and its surface C:O ratio were decreased, and that CWR-Cu group exhibited the largest decrease in both sustained-release carbon source density and C:O ratio and produced more -C=O, -COOH, and -OH, indicating that the sustained-release carbon source in CWR-Cu was most fully utilized. In summary, adding sustained-release carbon source to UVFCW improved the pollutant removal efficiency. Our findings provided theoretical basis and technical support for the application of actual projects.

Supplemental Information

Supplemental Information 1 Effluent water quality along with operation time.

Supplemental Information 2 Pollutant indicators at different depths.

Supplemental Information 3 The relative abundance proportion of phylum-level and genes-level microorganisms.

Supplemental Information 4 Infrared spectrum analysis of slow-release carbon source before and after use.

CWCu represents uniform slow-release carbon source, CWCa represents upper slow-release carbon source, CWCb represents lower slow-release carbon source, CWCK represents original slow-release carbon source.

Additional Information and Declarations

Competing Interests

Author Contributions

DNA Deposition

Data Availability

Yuechang Wang, Xiaoping Wang, Shaoming Cui and Gang Peng are employed by Beijing Further Tide Eco-construction Co., Ltd.

Shuang Liu, Yuanjun Su and Shuting Xie are employed by Further Tide Eco-construction (Hubei) Co., Ltd.

Yuechang Wang conceived and designed the experiments, performed the experiments, prepared figures and/or tables, and approved the final draft.

Lin-Lan Zhuang conceived and designed the experiments, authored or reviewed drafts of the article, and approved the final draft.

Shuang Liu performed the experiments, authored or reviewed drafts of the article, and approved the final draft.

Yuanjun Su analyzed the data, prepared figures and/or tables, and approved the final draft.

Zhen Hu conceived and designed the experiments, authored or reviewed drafts of the article, and approved the final draft.

Jian Zhang conceived and designed the experiments, authored or reviewed drafts of the article, and approved the final draft.

Xiaoping Wang performed the experiments, authored or reviewed drafts of the article, and approved the final draft.

Shaoming Cui analyzed the data, authored or reviewed drafts of the article, and approved the final draft.

Gang Peng conceived and designed the experiments, performed the experiments, prepared figures and/or tables, and approved the final draft.

Shuting Xie analyzed the data, prepared figures and/or tables, and approved the final draft.

The following information was supplied regarding the deposition of DNA sequences:

The group II intron/IEP sequences are available at GenBank: PRJNA1173921.

The following information was supplied regarding data availability:

The original data is available in the Supplemental Files.

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
