# Peer review of "Optimization and action mechanism of pollutant removal performance of unsaturated vertical flow constructed wetland (UVFCW) driven by substained-release carbon source"

_PeerJ, doi:10.7717/peerj.18819_

## Round 0.1 · original submission · Major Revisions

In this manuscript, the pollutant removal effect of constructed wetlands was optimized from the carbon and oxygen regulation perspective. Major revisions are needed - although the comments of both reviewers are brief, it is important to address them all in detail

Reviewer 1 ·

Basic reporting

In this manuscript, the pollutant removal effect of constructed wetlands was optimized from the perspective of carbon and oxygen regulation. A low-cost and long-term carbon source was modified, and three types of the carbon source distribution form were further analyzed through the performance of pollutant removal, the effective range of carbon release, and the microbial mechanism. Finally, an optimized layout including oxic/anaerobic ratio and carbon source distribution was proposed for CWs, which provided technical support for efficient water purification in CWs.

Experimental design

Well-designed experiment.

Validity of the findings

All underlying data have been provided; they are robust, statistically sound, & controlled.
The conclusions are well stated.

Additional comments

However, there are still some problems that need to be modified before final acceptance.
Specific recommendations are as follows:
(1) In Section 3.2, Lines 240-251, the removal of TN and NO3--N by the placement of slow-release carbon sources was not discussed in depth and could be further analyzed.
(2) In Lines 169-172, Please provide appropriate references.
(3) Adjust the size of the text in Fig. 7 (b) and keep them consistent within one figure.
(4) Check the superscript and subscript through the full text. For example, ' CWR-Cu ' in Lines 28, 189 and 305 should be changed to ' CWR-Cu '.
(5) Please check the citation format of full-text references, e.g., Lines 51-52, Lines 187-188, Line 278, and Lines 363-364.

·

Basic reporting

The article is very well written and with impactful research contributions in the field of COD, TN, and TP removal from wastewater. The use of UVFCW delivers excellent performance. However, the authors need to address few queries before the article can be accepted.
The authors need to justify the use of UVFCW? What do they mean by unsaturated and how it helps in creating the DO gradient needs to be justified.
The use of sustained release of carbon source needs to be justified.
Also, in the article it is written substained. Why is it so? It should be sustained right?

Experimental design

How is the influent concentration for COD, TP, ammonia are constant?
DO, ORP, and pH of influent and effluent and inside the reactor needs to be reported to establish nitrogen removal.
An actual figure of the reactor is required for the readers to better understand the configuration.
From Fig. 1 it is not clear how the carbon source is below or above the device? What do you mean by device?

Validity of the findings

No comment

Additional comments

No comment

---

## Round 0.2 · accepted · Accept

This revised version is suitable for publication in PeerJ.

Reviewer 1 ·

Basic reporting

The revised manuscript answered all my questions. I have no further comment.

Experimental design

The revised manuscript answered all my questions. I have no further comment.

Validity of the findings

The revised manuscript answered all my questions. I have no further comment.

·

Basic reporting

The paper may be accepted since the authors have responded to the comments in a suitable manner

Experimental design

No comment

Validity of the findings

No comment

Additional comments

No comment